# Less Animal-Based Food, Better Weight Status: Associations of the Restriction of Animal-Based Product Intake with Body-Mass-Index, Depressive Symptoms and Personality in the General Population

**DOI:** 10.3390/nu12051492

**Published:** 2020-05-20

**Authors:** Evelyn Medawar, Cornelia Enzenbach, Susanne Roehr, Arno Villringer, Steffi G. Riedel-Heller, A. Veronica Witte

**Affiliations:** 1Department of Neurology, Max Planck Institute for Human Cognitive and Brain Sciences, 04103 Leipzig, Germany; villringer@cbs.mpg.de (A.V.); witte@cbs.mpg.de (A.V.W.); 2Berlin School of Mind and Brain, Humboldt-Universität zu Berlin, 10117 Berlin, Germany; 3Center for Stroke Research Berlin (CSB), Charité Universitätsmedizin, Berlin, 10117 Berlin, Germany; 4Institute for Medical Informatics, Statistics and Epidemiology, University of Leipzig, 04107 Leipzig, Germany; cornelia.enzenbach@life.uni-leipzig.de; 5Institute of Social Medicine, Occupational Health and Public Health, University of Leipzig, 04109 Leipzig, Germany; susanne.roehr@medizin.uni-leipzig.de (S.R.); steffi.riedel-heller@medizin.uni-leipzig.de (S.G.R.-H.); 6Day Clinic for Cognitive Neurology, University Hospital, Leipzig University, 04103 Leipzig, Germany

**Keywords:** body weight, diet, plant-based, meat, depression, personality, population-based, cross-sectional

## Abstract

Restricting animal-based products from diet may exert beneficial effects on weight status; however, less is known about such a diet and emotional health. Moreover, personality traits, for example high neuroticism, may contribute to restrictive eating habits and potentially confound diet-health associations. We aim to systematically assess if restrictive dietary intake of animal-based products relates to lower weight and higher depressive symptoms, and if differences in personality traits play a significant role. Cross-sectional data from the baseline LIFE-Adult study were collected from 2011–2014 in Leipzig, Germany (*n* = 8943). Main outcomes of interest were dietary frequency of animal-derived products in the last year measured using a Food Frequency Questionnaire (FFQ), body-mass-index (BMI) (kg/m^2^), and the Center of Epidemiological Studies Depression Scale (CES-D). Personality traits were assessed in a subsample of n = 7906 using the Five Factor Inventory (NEO-FFI). Higher restriction of animal-based product intake was associated with a lower BMI, but not with depression scores. Personality, i.e., lower extraversion, was related to higher frequency of animal product intake. Moreover, personality traits were significantly associated with depressive symptoms, i.e., higher neuroticism, lower extraversion, lower agreeableness, lower conscientiousness, and with higher BMI. These findings encourage future longitudinal studies to test the efficacy of restricting animal-based products as a preventive and therapeutic strategy for overweight and obesity.

## 1. Introduction

Current debates assign animal product-restrictive eating patterns, such as vegetarian and vegan, either health benefits or risks [1]. For example, epidemiological studies such as the Adventist studies (*n* = 22,000–96,000) reported lower all-cause mortality rates and lower prevalence of cardiovascular diseases in participants with plant-based eating habits compared to those with omnivorous diets [2,3]. Other studies like the EPIC-Oxford study (*n*~64,000) [4] and the 45 and “Up Study” (*n*~267,000) [5], however, showed no effect of a plant-based diet on mortality rate. An 18-year follow-up analysis of the EPIC-Oxford study showed, on the one hand, a decrease of ischaemic heart disease prevalence, and, on the other hand, an increased odds ratio for total stroke, in fish-eaters and vegetarians compared to meat-eaters [6]. Intervention studies in small to moderate sample sizes (*n*~100) indicated that medium-term vegan diets (12–74 weeks), compared to omnivorous diets, lead to weight loss and to a decrease in type 2 diabetes symptoms, even when caloric intake was comparably low between the diets [7,8,9].

While the exact mechanisms mediating these effects are far from fully understood, improved energy metabolism, reductions of systemic low-grade inflammation and changes in microbiome-gut-brain signaling might play a pivotal role [1,10,11,12,13,14].

Further, individuals showing restrictive eating patterns, i.e., excluding animal-derived food, may be more or less prone to develop mood disturbances compared to those with omnivorous eating styles: large epidemiological studies (*n* = 6422–90,380) showed higher depressive symptoms in vegetarians and vegans [15,16,17] and in those with orthorexic behaviour [18]. Yet other (smaller) cross-sectional (n = 620) and interventional (*n* = 39–291) studies proposed a positive effect of plant-based diets on well-being and subclinical depression scores [19,20,21,22]. Recently, it has been suggested that, not meat-restriction per se, but the number of excluded food groups predicts higher depressive scores [17].

In addition, weight changes relate to depressive symptoms [23], and obesity and depression might share genetic pathways and personality traits, in particular neuroticism [24]. For example, studies showed that higher neuroticism and lower conscientiousness correlate with a higher BMI and more depressive symptoms [25,26]. Moreover, differences in personality traits and in demographic factors such as age, sex and education have been linked to more or less restrictive lifestyle habits, including diet [27,28,29].

Taken together, these factors likely introduce confounding in studies assessing the relationship between diet, weight status and depressive symptoms separately. However, previous studies have not always accounted for these complex dependencies, rendering a definitive conclusion difficult as to whether animal product-restrictive eating habits convey health benefits or risks. We therefore aimed to systematically determine the interplay between animal-restrictive vs. omnivorous dietary habits (measured on a continuum as frequency of animal-based food intake), weight status, depressive symptoms and personality traits in a large population-based sample of adults in Germany. 

We hypothesized that: (1) higher restriction of animal-based products is associated with lower BMI (kg/m^2^), even when accounting for potential confounding factors; (2) higher restriction of animal-based products is associated with certain personality traits, measured using the Five-Factor Inventory (NEO-FFI); (3) higher restriction of animal-based products is associated with higher depressive symptoms scores (measured using CES-D), yet the association may attenuate when taking differences in demographics and personality traits into account.

## 2. Materials and Methods

All analyses and hypotheses have been preregistered in the Open Science Framework (OSF) at https://osf.io/4w69q. Participants were drawn from the population-based Leipzig Research Centre for Civilization Diseases (LIFE)-Adult cohort, which aims to explore causes and developments of common civilization diseases such as obesity, depression and dementia [28] (Figure 1). Briefly, *n* > 9500 adult participants (“Adult Baseline”) were randomly selected based on sex and year of birth (age range 18–80 y, with a main proportion focus between 40–80 y), from the city registry of Leipzig, a major city with 550,000 inhabitants in the east of Germany. Additional volunteers (*n* > 900, randomly recruited from the city registry and from local databases, “Adult Baseline Plus”) were included for periods of feasibility testing, piloting and finalization. Data collection was conducted from August 2011 to November 2014 at a single site in cooperation of the Faculty of Medicine, Leipzig University and the Max Planck Institute for Human Cognitive and Brain Sciences. All participants underwent anthropometric measurements and answered extensive questionnaires regarding dietary habits, depressive mood and personality (see below for details). 

### 2.1. Inclusion Criteria

The initial dataset consisted of *n* = 10,083 participants. Subjects were included in the analysis based on standardized rules, if valid and complete measures of age, sex, education, BMI, CES-D and FFQ were available, resulting in a sample of *n* = 8943 (sample 1) and a subsample with additionally available personality trait measures of *n* = 7906 (sample 2, Figure 1). Note that results from sample 2 may slightly deviate from the previously reported pilot analyses in the OSF registration due to partially non-overlapping samples and an extension to a personality questionnaire that was widely available in the dataset.

### 2.2. Ethics

The institutional ethics board of the Medical Faculty of the University of Leipzig raised no concerns regarding the study protocol and all participants provided written informed consent. Code described in the manuscript will be made publicly and freely available without restriction at https://osf.io/m7hxk/?view_only=91863f44bae44371a1317072334df9fd.

### 2.3. Demographics

Education levels were computed according to Comparative Analysis of Social Mobility in Industrial Nations levels (CASMIN) [30] into three levels (low, middle, and high).

### 2.4. Anthropometry

Body weight was measured with scale SECA 701, height was measured with height rod SECA 220 (SECA GmbH & Co. KG, Hamburg, Germany). Body weight (kg) and body height (m) were used to calculate body-mass-index (BMI) (kg/m^2^). For additional analyses WHO classification for obesity was used: underweight < 18.5 kg/m^2^, normal-weight >= 18.5 and < 25 kg/m^2^, overweight >= 25 and < 30 kg/m^2^, obese >= 30 kg/m^2^.

### 2.5. Personality

Personality traits were measured with the German version of the Big Five via Short Forms (16-Adjective Measure) [31]; subscales were computed for Neuroticism, Extraversion, Openness, Agreeableness and Conscientiousness by building summed scores according to the test’s manual (higher scores indicate more pronounced traits, lower scores indicate less pronounced traits). In a subsample, personality traits were measured with the German version of the NEOFFI-30 [32,33].

### 2.6. Depressive Scores

Depressive scores (self-reported) were assessed by the Centre of Epidemiologic Studies-Depression (CES-D) scale [34]. Total CES-D score was calculated as a sum of responses to all 20 questions, with higher scores indicating more diverse and/or more frequent depressive symptoms.

### 2.7. Dietary Restriction Scores (DRS)

Food group items were taken from a questionnaire asking for self-reported food intake frequency over the last 12 months. A composite score for the restriction of animal-derived food items was calculated (Figure 2), including nine questions regarding the following food groups: meat, processed meat and cold cuts, fish, eggs, dairy (yoghurt and cream cheese), cheese, milk and butter (animal DRS). Answers ranged from multiple times daily (1 per item; 9 for summed score), daily/(almost) daily, multiple times a week, weekly, 2–3 times monthly, 1 or less a month to (almost) never (7 per item; 63 for summed score). The higher the score, the lower the frequency of consumption of animal-based products. Light products were recoded from 1–5 to 1–7, and either the normal or the light product was chosen for scoring depending on higher frequency; if both were equally frequent, the normal item was chosen (applicable for processed meat/cold cuts, dairy, cheese, butter and milk). Measures were ordinal, but for analysis purposes treated as linear, which is a common procedure for scoring lifestyle questionnaire data [35,36] and has been shown to perform robustly in parametric analyses [37]. Note that the questionnaire did not include an option such as “I prefer not to answer” or “I don’t know”. Missing values were replaced by the population mean in line with recommendation to use imputation for missing values in nutritional epidemiology [38]. Subjects with >20% of missing answers out of the 33 food items (excl. drink items) were excluded from the analysis (code and supplementary info available here (https://osf.io/m7hxk/?view_only=91863f44bae44371a1317072334df9fd).

To further investigate the difference between leaving out primary (meat, bone, and marrow, representing meat-restrictive diets) and/or secondary (stemming from animal labor, e.g., milk, representing vegetarian diets) animal products from the diet, we further tested whether potential associations were specific to either food groups by computing two additional scores: (a) primary DRS and (b) secondary DRS (Appendix A).

An additional score represents the number of restricted food items (adapted from [17] by counting all *(almost) never* of 33 items in the FFQ (excluding drinks and light products) (score min. 0 to max. 33) within the last 12 months (5.1 ± 2.9 items (mean ± SD), range 0–19) (overall DRS).

All computed scores were normally distributed (skewness < 1.0, kurtosis <= 2.0) (Appendix A). Moderate positive correlations were observed between meat and cold cuts (ρ = 0.46), processed meat and meat (ρ = 0.26), processed meat and cold cuts (ρ = 0.22), dairy and cheese (ρ = 0.42), and dairy and milk (ρ = 0.28) consumption (Appendix A). A correlation matrix of all key variables of interest, including restrictive dietary patterns, BMI, depressive symptoms and personality traits is available (Appendix A).

### 2.8. Statistical Models

The main analysis included linear regression models to examine the association of animal DRS and BMI (model 1), depressive symptoms (model 3) and personality traits in a multivariate analysis of covariance (MANCOVA) (model 2). More specifically, model 1 tested whether animal DRS predicted BMI, adjusting for age, sex and education. Model 2 tested whether animal DRS (factor) was associated with the different personality traits (five subscales of the NEO-FFI as dependent variables), accounting for age, sex and education (covariates). Model 3 tested whether animal DRS predicted CES-D when accounting for age, sex and education; and, additionally, accounted for personality factors and BMI. All variables were normally distributed (skewness < |1.06|, kurtosis < |2.08|), personality traits (skewness < |1.05|, kurtosis < |3.2|), except for CES-D (skewness 1.4, kurtosis = 3.3), which was therefore log-transformed (log 10(CES-D+1). Analyses were computed in R version 3.6.1 using lm, lm.beta and ggplot2 for visualization. Statistical significance was set at alpha = 0.05/3 = 0.015 in the main analyses to adjust for multiple testing with the Bonferroni method and at *p* < 0.05 in all additional analyses.

## 3. Results

We included 8943 subjects for analyses regarding diet, BMI and depressive symptoms (see Table 1 for demographics), and 7906 participants in sample 2 for the subsample analysis additionally investigating personality traits (see Table 2). Due to the focus on the age range between 40–80 years in the selection of participants drawn from the city registry [28], the studied sample was on average middle-aged (mean age 57 y) and showed a skewed adult age distribution to the right. In addition, the study included slightly more women than men (4609F, 4334M) and a very wide BMI range (16–57 kg/m^2^, on average 27.3 kg/m^2^). About 2% of the sample (*n* = 237) reported to have adopted an exclusively vegetarian diet at least once throughout their life, and about 5% (*n* = 547) a mainly vegetarian diet.

Linear regression models detected that lower animal DRS, i.e., higher frequency of animal-based products consumption, related to higher BMI in sample 1 (*n* = 8943; β = −0.07, *p* < 0.001, Bonferroni-corrected), corrected for confounders (age, sex, education). Higher age, being male and lower education were also significantly associated with higher BMI, with the four factors together explaining about 6% of the variance in BMI (overall model adj. R^2^ = 0.06, *p* < 0.001, Bonferroni-corrected) (Figure 3A, Table 3). Here, age showed the steepest slope (*n* = 8943; β = 0.08, *p* < 0.001; Figure 3B). Similar results emerged when restricting the analysis to the smaller sample 2 (data not shown). When additionally correcting for personality traits the association between BMI and animal DRS remains significant (*n* = 7906; β = −0.07, *p* < 0.001, Bonferroni-corrected), further certain personality traits show significant associations with BMI (neuroticism: β = −0.05, *p* < 0.001; openness: β = −0.05, *p* < 0.02; agreeableness: β = 0.13, *p* < 0.001; conscientiousness: β = −0.2, *p* < 0.001; all *n* = 7906) (Table 3).

Further, in sample 2 we found a significant association between frequency of animal-based products and personality traits, when correcting for age, sex and education (n = 7906; MANCOVA, F_(5,7897)_ = 2.8, *p* < 0.02): higher restriction of animal products was negatively associated with extraversion (F_(1,7897)_ = 9.8, *p* = 0.002) (Figure 4, Table 4). Although non-significant, animal DRS was positively associated with neuroticism (F_(1,7897)_ = 3.5, *p* = 0.06) and negatively with openness (F_(1,7897)_ = 3.4, *p* = 0.07). Likewise, sex was significantly associated with all five personality traits; and age and education with four of them (all except for agreeableness) (Table 4).

Lastly, frequency of animal-based products did not predict variance in depressive symptoms in sample 1 (*n* = 8943, β = 0.001, *p* = 0.12), according to a linear regression model (model 3) that corrected for age, sex, and education (overall model: R^2^ = 0.04, *p* < 0.001, Bonferroni-corrected) (Table 5). This was also the case for sample 2 (*n* = 7906, animal DRS: β = 0.001, *p* = 0.10; overall model; R^2^ = 0.04; *p* < 0.001), also when additionally correcting for personality traits and BMI (n = 7906, animal DRS: β = 0.013, *p* = 0.2; overall model; R^2^ = 0.21; *p* < 0.001) (Table 5). Instead, higher neuroticism (β = 0.4, *p* < 0.001), lower extraversion (β = −0.08, *p* < 0.001), lower openness (β = −0.07, *p* < 0.001), lower conscientiousness (β = −0.08, *p* < 0.001) and higher BMI (β = 0.06, *p* < 0.001) correlated with depressive symptoms (overall model explaining 21% of variance on depressive symptoms score) (Figure 5, Table 5). 

To confirm whether results were not driven by extreme cases with pathological underweight, we excluded underweight individuals (BMI <= 18.5 kg/m^2^) from the analysis (n = 51, 17.8 ± 0.6 kg/m^2^ (mean ± SD), range 16–18.5). This did not change the results from the main analyses (data not shown).

### Ancillary Analyses

Restricting primary animal source products (i.e., (processed) meat, cold cuts) was significantly associated with a lower BMI (n = 8943; β = −0.25, *p* < 0.001, Figure 6), but not restricting intake of secondary animal products (cheese, milk, eggs) (n = 8943, β = −0.02, *p* = 0.16) (Table 6). Note the somewhat stronger association of primary animal-based products with BMI compared to the “comprehensive” animal-product DRS score, resulting in a more negative β coefficient.

Investigating differences in personality, higher primary animal DRS was significantly associated with lower neuroticism (F_(1,7897)_ = 27.5, *p* < 0.001), higher openness (F_(1,7897)_ = 45.1, *p* < 0.001), higher agreeableness (F_(1,7897)_ = 262.5, *p* < 0.001) and higher conscientiousness (F_(1,7897)_ = 63.1, *p* < 0.001). Higher secondary animal DRS was significantly associated with lower extraversion (F_(1,7897)_ = 11.1, *p* < 0.001), lower openness (F_(1,7897)_ = 26.9, *p* < 0.001), lower agreeableness (F_(1,7897)_ = 106.7, *p* < 0.001) and lower conscientiousness (F_(1,7897)_ = 14.2, *p* < 0.001) (all: n = 7906, MANCOVA, corrected for age, sex and education) (Appendix A). 

In contrast to the comprehensive animal product DRS, the scores displaying restriction of either primary or secondary origin animal products were also associated with lower and higher depressive symptoms scores, respectively (n = 8943, primary animal-product DRS: β = −0.003, *p* = 0.04; secondary animal-product DRS: β = 0.002, *p* = 0.02; models adjusted for age, sex and education). These divergent associations, however, failed to reach significance when additionally correcting for personality traits (n = 7906, all |β| < 0.002, all *p* > 0.10, adjusted for age, sex, education and personality) (Table 7).

Further, we found a strong positive correlation between the frequency of animal-based products (animal DRS) and the number of restricted food groups considering all 33 items (overall DRS) (ρ(8941) = 0.52, *p* < 0.001) (Figure 7A).

Considering the number of restrictive food items in general, we found that a higher score of total excluded food items related to lower BMI (sample 1: β = −0.15, t = −8.8, *p* < 0.001, R^2^ = 0.07, corrected for age, sex and education) (Figure 7B, Table 6).

The number of restricted food items was significantly associated with lower agreeableness (F_(1,7897)_ = 15.7, *p* < 0.001) and higher conscientiousness (F_(1,7897)_ = 53.9, *p* < 0.001) (*n* = 7906, MANCOVA, F_(5,7897)_ = 11.8, *p* = < 0.001, for model comparison against null model, corrected for age, sex and education) (Table 8).

Surprisingly, a higher number of restricted food items was weakly yet significantly associated with lower depressive symptoms scores (β = −0.004, t = −4.1, *p* < 0.001, R^2^ = 0.05, corrected for age, sex and education) (similar results in sample 2 (data not shown)), also when additionally correcting for differences in personality traits (β = −0.003, t = −2.7, *p* < 0.007, R^2^ = 0.21) (Figure 7C, Table 9).

## 4. Discussion

In this large cross-sectional analysis of ~9000 individuals from the general population, lower frequency of eating animal-based products was significantly associated with lower BMI, even when adjusting for confounding effects of age, sex and education. No significant associations emerged between animal-based products consumption and depressive symptom scores when taking personality into account. Frequency of animal-based product consumption was associated with personality traits, in particular with lower extraversion. Surprisingly, not diet but personality was significantly associated with depressive mood.

While the selection of our mid- to older age adult urban sample from Eastern Germany depended on the city registry of Leipzig, participants with a low social status and an unhealthy lifestyle were somewhat underrepresented compared to the general population [39]. However, the BMI range can be considered representative of a German population of this age.

### 4.1. Weight Status

Our finding that eating meat and dairy products less frequently relates to lower BMI is in line with some, but not all, epidemiological and moderate-term randomized interventional trials which point in this direction too [1,40,41]. In addition, results remained stable even after adjusting for education, which is a strong predictor of both obesity [42] and eating habits [43], and when taking inter-individual variance in personality traits into account [44]. Speculating on possible underlying mechanisms, animal-derived products in general are often denser in calories and in total and saturated fats compared to plant-based foods [45]. In addition, meat and dairy products are oftentimes consumed as processed food, e.g., processed meat, cold cuts, deep-fried meat/fish or high-processed snack products, further augmenting their caloric footprint. Thus, lower caloric intake might underlie the observed link between lower frequency of animal-based product consumption and lower BMI. Moreover, recent observations of changes in the gut microbiome due to diet raise the hypothesis that a different distribution of gut bacteria in plant-based dieters alters the ingestion rate of calories from food [46], thereby further limiting caloric intake (or bioavailability). However, while these causal pathways between lower frequency of animal-based product intake leading to lower or sustained body weight seem biologically plausible, the association between lower animal-based product intake and lower weight in our cohort might also be a result of lower body weight leading to less animal-based product intake or unknown shared factors that modulate both weight and diet. Future longitudinal observations and interventional trials are needed to further test the above-described hypothesis or its alternatives.

The positive association between restriction of meat products on weight status and the lack of a significant correlation for secondary animal products found in this study and previously by others [47,48,49] could possibly be explained by a higher proportion of highly processed meat items, leading to higher net energy intake and potentially to higher caloric intake [50]. Further, ongoing discussions on motivations for following certain diets support the view that restraint eating is not directly linked to vegetarian or vegan diets but more common in flexitarians who restrict meat intake with the goal of weight control, which in contrast is not the most common driver in plant-based dieters [51]. 

While, due to the cross-sectional design using self-reported FFQ data, estimates of absolute numbers of the strength of the association between diet and BMI are difficult, our findings may be relevant for public health. Considering that changing a conventional Western omnivorous dietary habit to a more plant-based diet, i.e., avoiding (processed) meat and cold cuts and limiting dairy, cheese and egg intake, would lead to an increase in animal DRS of 20 points, this would translate into ~1.2 kg/m^2^ lower BMI. For someone with a frequent intake of primary and secondary animal-product intake (low animal DRS) this could mean, for example, reducing all animal-based products from multiple times a day to multiple times a week (“flexitarian diet”) or excluding some animal items altogether (“vegan” or “vegetarian” diet). For a 175 cm tall human this would translate into 4 kg of body weight. If obese (e.g., 100 kg, i.e., BMI = 32.7 kg/m²), this would mean a reduction of 4% body weight; if overweight (e.g., 80 kg, BMI = 26.1 kg/m²) this would mean a reduction of 5% body weight. As a reduction of 5–10% body weight has been shown to significantly reduce obesity-associated co-morbidities in overweight and obesity [52,53,54,55,56,57,58], restricting dietary intake of animal-based products may be one way to achieve this weight loss goal, and may help to reduce the societal burden of obesity-related diseases and environmental impact caused by high animal-product diets [40]. However, these calculations have to be interpreted with caution, as our findings rely on self-reported and cross-sectional data only, and we could not quantify dietary intake with regard to the consumed total amounts of food. Future longitudinal observations and interventional trials are needed.

### 4.2. Depressive Symptoms & Personality Traits

In contrast to previous large cross-sectional studies [16,17] and a prospective study in patients with inflammatory bowel disease [59], frequency of animal-derived product consumption did not explain variance in depressive symptoms scores in the current sample.

However, intervention studies showed that a plant-based vegan diet compared to a conventional omnivorous diet reduced anxiety and depression or emotional distress [19,20,21,22], proposing that restricting animal-based products per se may not affect mental health, but rather exert beneficial effects. Notably, we observed that different personality traits and BMI predicted depressive symptom score, which hints towards shared neurobiological mechanisms with obesity [23,25]. These shared mechanisms might help to explain previous inconsistent findings of a proposed link between restrictive diets and depression: certain personality traits may increase the probability of restricting certain food groups from diet, such as openness and conscientiousness [60]. Such a correlative link between personality and restrictive eating, although missing in the current data, would thus also apply to restricting animal-based products and may explain higher depressive symptoms in vegetarians or vegans [16]. Moreover, sociological studies show that animal-restricted dieters are often stereotyped with a multitude of biases: detrimental health effects, restrictive lifestyle, sentimentalism, extremism, lower perceived masculinity [61,62,63]. Aversion to plant-based dieters could lead to higher social exclusion and depressive symptoms as a result. However, more longitudinal studies tracking newly transformed dieters are needed to clarify if avoiding animal-derived products affects mental health.

Differences in our results compared to previous evidence on personality differences in vegetarians may be due to demographic and societal environmental factors. Personality trait differences in vegetarians were found in a cohort of college students [15], which might be different to our sample of the general population, in terms of beliefs, motivation of dietary habits, etc. Geographical or cultural settings may also influence differences in the results such as westernized (USA [15], Germany (this study)) versus mainly-vegetarian Indian cohorts [29], who showed higher conscientiousness. Lastly, the popularity and availability of plant-based dishes is a strong modulator of societal acceptance and demand for those kinds of diets. For instance, increasing the offer from one to two plant-based meals in canteens led to an increase of 40–80% of plant-based meal purchases, underlining the importance of availability as a strong driver [64]. Since the interest in plant-based diets has been changing dynamically in the last decade, researches should take period and location into account when comparing studies. 

Strengths of our study comprise the large, well-characterized population based cohort enabling us to carefully control for important confounders such as education and personality. Moreover, recent studies and meta-analyses focused specifically on intake of red and processed meat and related health outcomes [65]. However the distinction of restricting diets to not consuming primary (vegetarian) and/or secondary animal-products (vegan) is oftentimes overlooked and therefore a strength of our study.

Our results further highlight a significant association of demographic variables with BMI, personality and depressive symptoms. This shows how individual factors such as age, sex and education are tightly linked to health (dis)advantages in our societies, and future studies on public health interventions should focus on those at particular risk, e.g., older males with lower education in the case of BMI, and older females with lower education in the case of depressive symptoms. In parallel, the cross-sectional nature of our analysis does not allow us to imply causal relations, therefore future longitudinal and experimental interventional studies need to test to what extent modifiable factors, such as education, could causally reduce obesity and depression, and how dietary strategies such as reducing animal-based products might help to mediate these effects.

### 4.3. Limitations

Firstly, limitations of our study include that the results are based on a cross-sectional study design and therefore cannot explain underlying causalities.

Secondly, our analyses are based on self-reported dietary food record, which does not reflect actual food intake; however, test-retest reliability is generally of good quality [66]. The FFQ used in this study has been adopted from the frequently used German National Health Interview and Examination Survey 1998 [67], which has, however, not been validated, at least to our knowledge. Moreover, the FFQ used did not ask for quantity of food intake, which limits the interpretability of the observed effects (for further discussion on possible mechanisms see [1]) as important confounders such as total kcal intake could not be considered in this analysis. Yet, beside this possible inaccuracy of self-reported food intake, we propose that excluding certain food groups for a timeframe of twelve months is presumably a strong and reliable indicator of actual food intake and exclusion of certain food groups. 

Thirdly, this study verifies depressive mood or depressive symptoms and not depression. The clinical interview remains the gold standard for identifying depression.

Fourthly, we did not account for ethnicity as these statistics were not available for this cohort. Based on historical data on immigration for this region, we estimate that the majority of the LIFE cohort were probably of Caucasian ethnicity and <1% were non-Caucasian.

Lastly, as frequently found in nutritional epidemiology, in our analysis socioeconomic status was accounted for by level of education, not income and occupational status [68]. This one-dimensional analysis might result in limited generalizability of the results. However, education can be viewed as a more long-term indicator of socioeconomic status compared to more dynamic monthly net income (as provided in this dataset).

## 5. Conclusions

Taken together, using a large cross-sectional analysis we observed that a lower frequency of animal-based products was related to lower BMI, while no link between animal-based products intake and depressive symptoms scores emerged. Thus, our findings may suggest that a lower frequency of animal-based products could be able to convey benefits on weights status, hinting to the capacity of plant-based diets as a potentially relevant target for the intervention of obesity and overweight, in particular by reducing the frequency (and probably the amount) of (especially primary source) animal-based products. Long-term interventional trials are needed to test this hypothesis and to clarify the underlying mechanisms.

## Figures and Tables

**Figure 1 nutrients-12-01492-f001:**
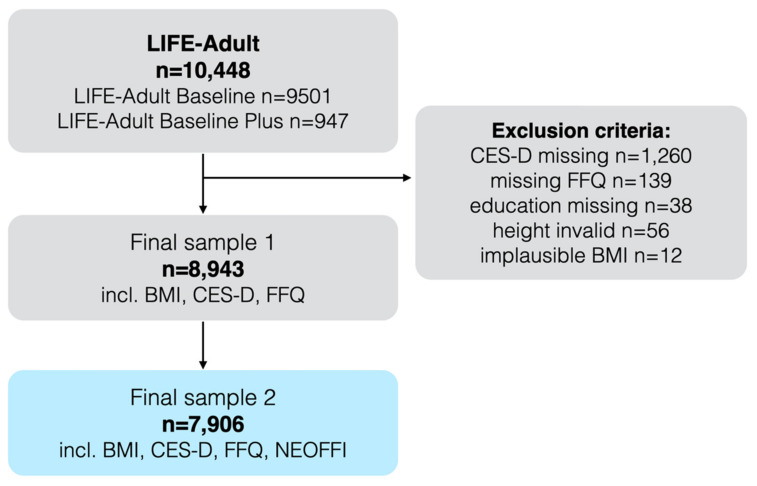
Flowchart of sample selection for sample 1 and sample 2. Abbreviations: BMI = Body-Mass-Index, CES-D = Center of Epidemiological Studies Depression Scale, FFQ = Food Frequency Questionnaire, NEOFFI=NEO Five-Factor-Inventory.

**Figure 2 nutrients-12-01492-f002:**
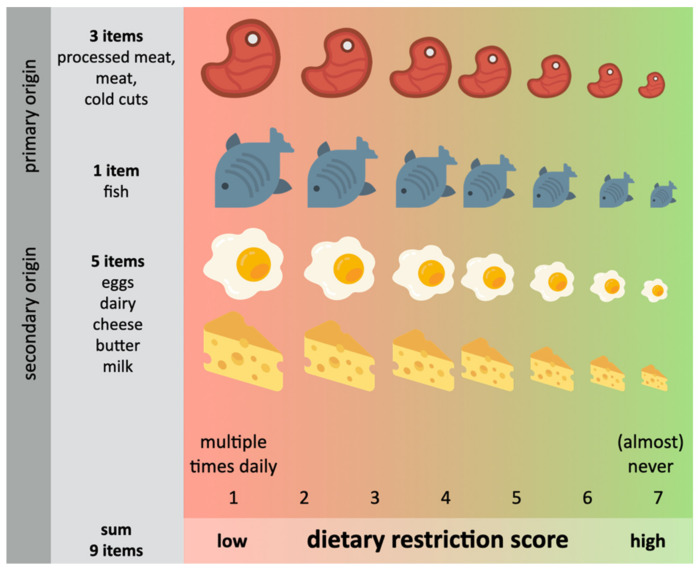
Concept of dietary restriction score (DRS) based on the frequency of consumption of animal-based products over the last 12 months based on nine items from the FFQ. Copyright icons: all icons by Smashicons.

**Figure 3 nutrients-12-01492-f003:**
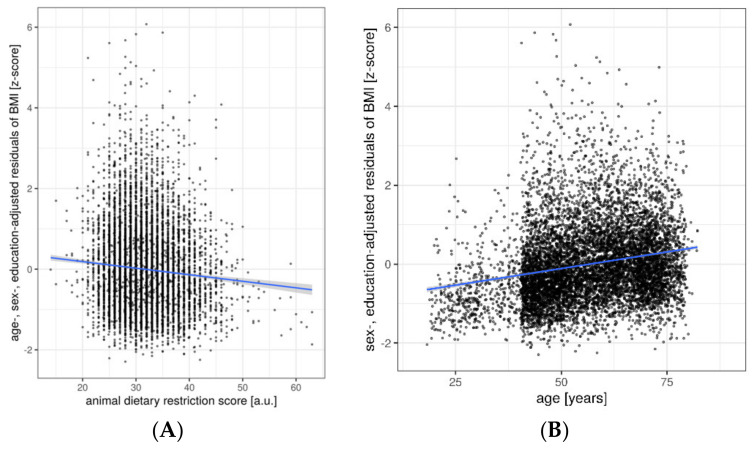
Association between body-mass-index (BMI) and demographic and lifestyle factors (**A**) animal DRS (**B**) age, residuals plotted according to regression model 1 (sample 1 *n* = 8943). Line gives regression fit. Point size = 1. Abbreviations: a.u. = arbitrary units.

**Figure 4 nutrients-12-01492-f004:**
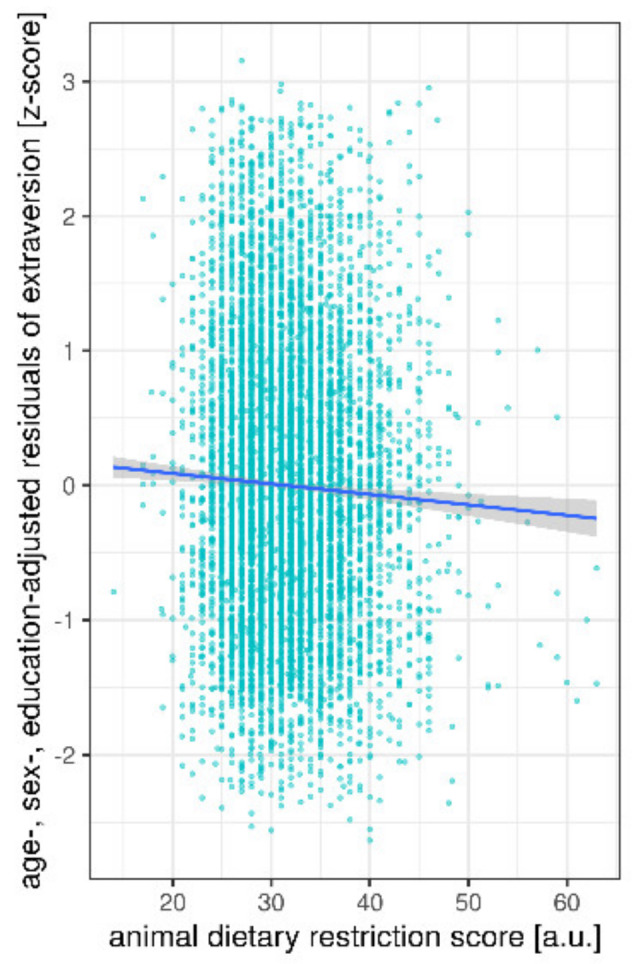
Association between animal DRS and extraversion, residuals plotted according to regression model 2 (sample 1 *n* = 8943). Line gives regression fit. Point size = 1. Abbreviations: a.u. = arbitrary units.

**Figure 5 nutrients-12-01492-f005:**
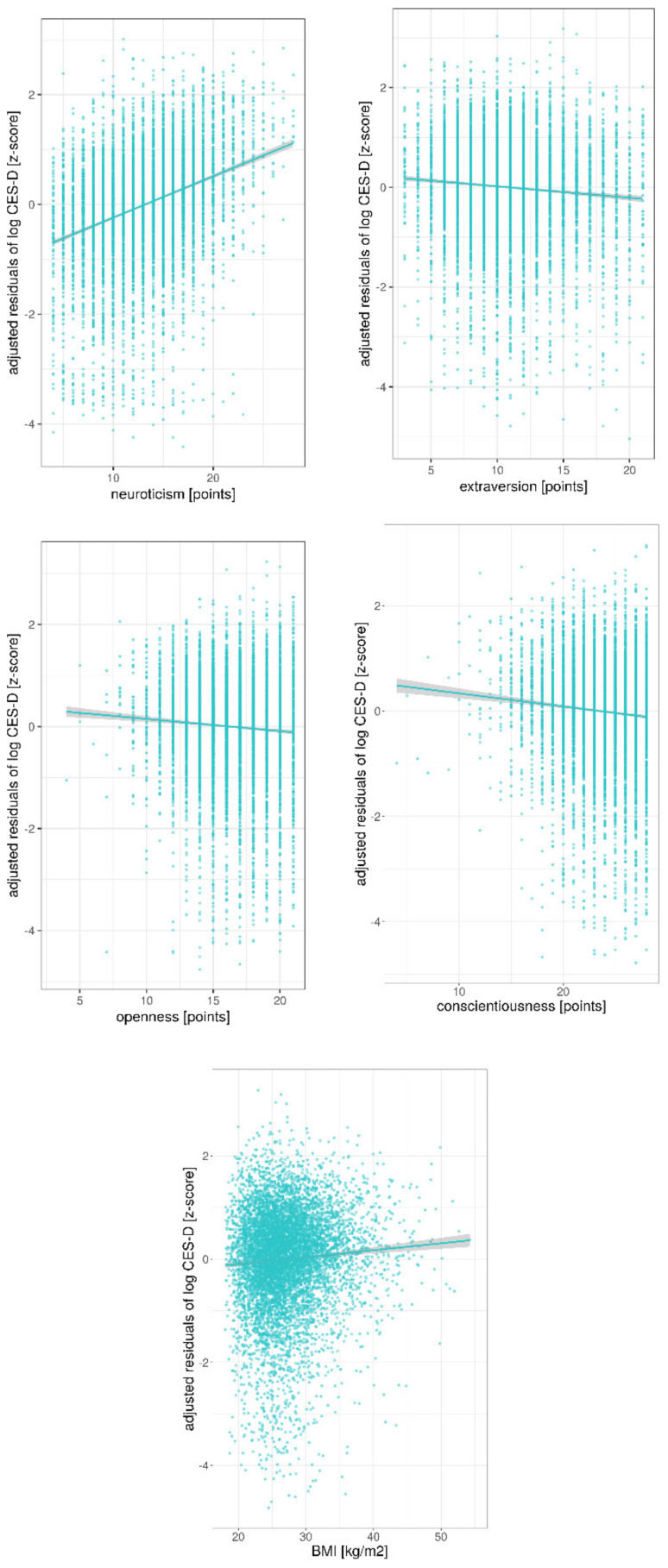
Significant association between personality traits and depressive symptoms in sample 2 (*n* = 7906) corrected for age, sex, education, animal DRS and the respective four other subscales of personality for neuroticism, extraversion, agreeableness, conscientiousness and BMI. Lines give regression fit. Position size = 2 (for personality) and 1 (BMI).

**Figure 6 nutrients-12-01492-f006:**
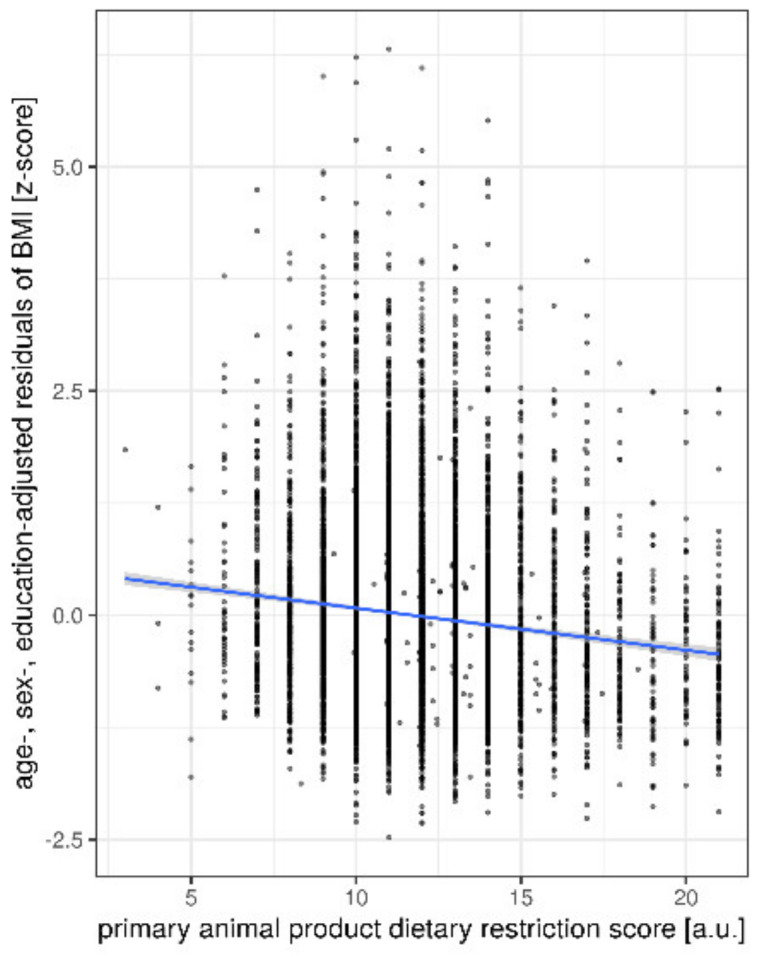
Restrictive animal-based product intake associated with lower BMI. Lines give regression fit. Position size = 1. Abbreviations: a.u. = arbitrary units.

**Figure 7 nutrients-12-01492-f007:**
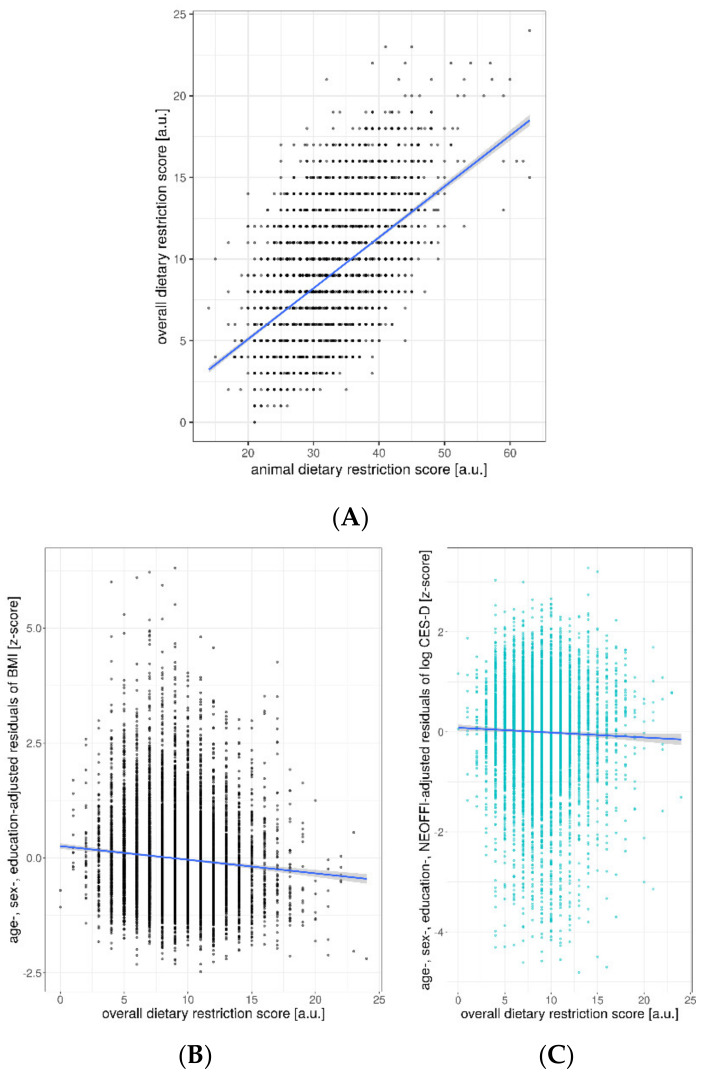
(**A**) Positive association between decreasing frequency of animal-based products and number of excluded food groups. Negative association between overall dietary restriction score with (**B**) BMI and (**C**) CES-D. Position size = 1. Abbreviations: a.u. = arbitrary units. Significant associations (p-values < 0.05) are indicated in bold.

**Table 1 nutrients-12-01492-t001:** Demographic characteristics for sample 1 and sample 2.

		Age (Years)	Sex	Education (CASMIN Levels)	Animal DRS (9–63)	BMI (kg/m^2^)	CES-D (0–60)
**Sample 1** **(*n* = 8943)**	**mean**	56.6	8943	2.28	31.53	27.25	10.69
		(18–82)	(4609F)	(1–3)	(14–63)	(16.2–57.3)	(0–53)
	**SD**	12.5	-	0.6	5.1	4.9	6.9
**Sample 2** **(*n* = 7906)**	**mean**	55.7	7906	2.31	31.55	27.16	10.57
		(18–82)	(4010F)	(1–3)	(14–63)	(16.2–57.3)	(0–53)
	**SD**	12.4	-	0.6	5.1	4.7	6.9

**Table 2 nutrients-12-01492-t002:** Personality traits according to the five factor personality questionnaire NEO-FFI (16 items) for sample 2 (*n* = 7906).

		Neuroticism	Extraversion	Openness	Agreeable-Ness	Conscientious-Ness
**Sample 2** **(*n* = 7906)**	**mean**	13.2	10.9	16.3	11.7	23.6
		(4–28)	(3–21)	(4–21)	(2–14)	(4–28)
	**SD**	4.4	3.7	2.7	2.0	3.2

**Table 3 nutrients-12-01492-t003:** Multiple regression analyses predicting BMI as function of age, sex, education and frequency of animal-based products (*n* = 8943).

	Adj. R^2^	B	C.I.	Beta	*p*
**BMI (model 1)**					
Model	0.06				<0.001
**sex**		−0.59	[−0.79 −0.40]	−0.06	**<0.001**
**education**		−0.67	[−0.83 −0.50]	−0.08	**<0.001**
**age**		0.08	[0.07 0.09]	0.21	**<0.001**
**animal DRS**		−0.07	[−0.09 −0.05]	−0.06	**<0.001**
**BMI (model 1)—sample 2 (df = 7896), corrected for personality**					
Model	0.08				<0.001
**sex**		−0.55	[−0.78 −0.33]	−0.06	**<0.001**
**education**		−0.65	[−0.83 −0.47]	−0.08	**<0.001**
**age**		0.09	[0.09 0.10]	0.24	**<0.001**
**animal DRS**		−0.07	[−0.09 −0.05]	−0.07	**<0.001**
**neuroticism**		−0.05	[−0.08 −0.03]	−0.05	**0.001**
**extraversion**		0.01	[−0.02 0.04]	0.01	0.42
**openness**		−0.05	[−0.10 −0.01]	−0.03	**0.01**
**agreeableness**		0.13	[0.07 0.19]	0.05	**<0.001**
**conscientiousness**		−0.20	[−0.23 −0.16]	−0.13	**<0.001**

B/beta represent unstandardized/standardized regression coefficients. Abbreviations: BMI = body-mass-index, DRS = dietary restriction score. Significant associations (p-values < 0.05) are indicated in bold.

**Table 4 nutrients-12-01492-t004:** Multivariate analysis of covariance (MANCOVA) analysis of animal DRS, age, sex, education on personality (*n* = 7906).

	Pillai’s Trace	F	df	num df	den df	*p*
**NEOFFI (model 2) (all factors, corrected for age, sex, education)**						
**sex**	0.17	322.2	1	5	7897	**<0.001**
**education**	0.04	66.9	1	5	7897	**<0.001**
**age**	0.04	69.3	1	5	7897	**<0.001**
**animal DRS**	0.002	2.8	1	5	7897	**0.016**
**NEOFFI Neuroticism**						
**sex**		327.6	1	5	7897	**<0.001**
**education**		113.5	1	5	7897	**<0.001**
**age**		28.5	1	5	7897	**<0.001**
animal DRS		3.5	1	5	7897	0.06
**NEOFFI Extraversion**						
**sex**		15.9	1	5	7897	**<0.001**
**education**		71.1	1	5	7897	**<0.001**
**age**		152.7	1	5	7897	**<0.001**
**animal DRS**		9.8	1	5	7897	**0.002**
**NEOFFI Openness**						
**sex**		7.3	1	5	7897	**0.007**
**education**		208.4	1	5	7897	**<0.001**
**age**		4.6	1	5	7897	**0.03**
animal DRS		3.4	1	5	7897	0.07
**NEOFFI Agreeableness**						
**sex**		953.5	1	5	7897	**<0.001**
**education**		1.0	1	5	7897	0.33
**age**		0.7	1	5	7897	0.39
**animal DRS**		0.03	1	5	7897	0.87
**NEOFFI Conscientiousness**						
**sex**		137.4	1	5	7897	**<0.001**
**education**		10.7	1	5	7897	**0.001**
**age**		148.4	1	5	7897	**<0.001**
**animal DRS**		0.0006	1	5	7897	0.98

Abbreviations: DRS = dietary restriction score. Significant associations (p-values < 0.05) are indicated in bold.

**Table 5 nutrients-12-01492-t005:** Multiple regression analyses predicting CES-D as a function of age, sex, education animal DRS (sample 1, *n* = 8493) and additionally personality traits (sample 2, *n* = 7906) and BMI.

	Adj. R^2^	B	C.I.	Beta	p
**CES-D (model 3)—sample 1 (df = 8938)**					
**Model**	0.04				<0.001
**sex**		0.04	[0.029 0.051]	0.071	**<0.001**
**education**		−0.09	[−0.10 −0.08]	−0.184	**<0.001**
**age**		0.001	[0.0007 0.0016]	0.050	**<0.001**
**animal DRS**		0.001	[−0.0002 0.0020]	0.016	0.12
**CES-D (model 3)—sample 2 (df = 7901)**					
**Model**	0.04				
**sex**		0.04	[0.0273 0.0523]	0.069	**<0.001**
**education**		−0.09	[−0.1001 −0.0786]	−0.180	**<0.001**
**age**		0.001	[0.0006 0.0016]	0.049	**<0.001**
**animal DRS**		0.001	[−0.0002 0.0022]	0.018	0.10
**CES-D (model 3)—sample 2 (df = 7896), corrected for personality**					
**Model**	0.21				
**sex**		0.011	[−0.001 0.024]	0.02	0.08
**education**		−0.06	[−0.07 −0.05]	−0.12	**<0.001**
**age**		0.0006	[0.0001 0.0011]	0.03	**0.015**
**animal DRS**		0.0005	[−0.0006 0.0015]	0.009	0.40
**neuroticism**		0.024	[0.022 0.025]	0.36	**<0.001**
**extraversion**		−0.006	[−0.008 −0.005]	−0.08	**<0.001**
**openness**		−0.007	[−0.010 −0.005]	−0.07	**<0.001**
**agreeableness**		−0.0004	[−0.004 0.003]	−0.003	0.80
**conscientiousness**		−0.008	[−0.009 −0.006]	−0.08	**<0.001**
**CES-D (model 3)—sample 2 (df = 7895), corrected for personality and BMI**					
Model	0.21				<0.001
**sex**		0.013	[0.0008 0.026]	0.02	**0.04**
**education**		−0.06	[−0.082 −0.039]	−0.11	**<0.001**
**age**		0.0002	[−0.066 −0.046]	0.01	0.32
**animal DRS**		0.001	[−0.004 0.002]	0.013	0.20
**neuroticism**		0.024	[0.022 0.025]	0.36	**<0.001**
**extraversion**		−0.006	[−0.008 −0.005]	−0.08	**<0.001**
**openness**		−0.007	[−0.010 −0.005]	−0.07	**0.14**
**agreeableness**		−0.0009	[−0.004 0.003]	−0.006	0.60
**conscientiousness**		−0.007	[−0.009 −0.005]	−0.08	**<0.001**
**BMI**		0.004	[0.002 0.005]	0.06	**<0.001**

B/beta represent unstandardized/standardized regression coefficients. Abbreviations: BMI = body-mass-index, CES-D = depressive symptoms scale; DRS = dietary restriction score. Significant associations (p-values < 0.05) are indicated in bold.

**Table 6 nutrients-12-01492-t006:** Multiple regression analyses predicting BMI as a function of age, sex, education and restriction of different dietary items (sample 1, n = 8493).

	Adj. R2	B	C.I.	Beta	p
**BMI (model 1)—primary animal DRS**					
Model	0.07				<0.001
sex		−0.18	[−0.38 0.03]	−0.018	0.10
**education**		−0.61	[−0.76 −0.44]	−0.074	**<0.001**
**age**		0.09	[0.08 0.10]	0.225	**<0.001**
**primary animal DRS**		−0.25	[−0.29 −0.21]	−0.132	**<0.001**
**BMI (model 1)—secondary animal DRS**					
Model	0.06				<0.001
**sex**		−0.63	[−0.84 −0.43]	−0.065	**<0.001**
**education**		−0.65	[−0.82 −0.49]	−0.079	**<0.001**
**age**		0.08	[0.07 0.09]	0.209	**<0.001**
secondary animal DRS		−0.02	[−0.04 −0.01]	−0.015	0.16
**BMI (model 1)—overall DRS**					
Model	0.07				<0.001
**sex**		−0.50	[−0.69 −0.30]	−0.051	**<0.001**
**education**		−0.70	[−0.83 −0.49]	−0.080	**<0.001**
**age**		0.09	[0.08 0.10]	0.221	**<0.001**
**overall DRS**		−0.15	[−0.18 −0.11]	−0.091	**<0.001**

B/beta represent unstandardized/standardized regression coefficients Abbreviations: BMI = body-mass-index, DRS = dietary restriction score. Significant associations (p-values < 0.05) are indicated in bold.

**Table 7 nutrients-12-01492-t007:** Multiple regression analyses predicting CES-D as a function of age, sex, education and primary and secondary dietary restriction score (sample 1 *n* = 8943, sample 2 *n* = 7906).

	Adj. R^2^	B	C.I.	Beta	p
**CES-D—sample 1 (df = 8938)**					
**Model**	0.04				**<0.001**
**sex**		0.05	[0.031 0.058]	0.08	**<0.001**
**education**		−0.09	[−0.100 −0.078]	−0.18	**<0.001**
**age**		0.001	[0.0007 0.0017]	0.05	**<0.001**
**primary DRS**		−0.003	[−0.005 −0.00008]	−0.02	**0.04**
**CES-D—sample 2 (df = 7896), corrected for personality**					
**Model**	0.21				<0.001
**sex**		0.014	[0.0008 0.0270]	0.02	**0.04**
**education**		−0.06	[−0.068 −0.048]	−0.12	**<0.001**
**age**		0.0006	[0.0001 0.0011]	0.03	**0.01**
primary DRS		−0.002	[−0.004 −0.001]	−0.01	0.21
**neuroticism**		0.024	[0.022 0.025]	0.36	**<0.001**
**extraversion**		−0.006	[−0.008 −0.005]	−0.08	**<0.001**
**openness**		−0.007	[−0.010 −0.005]	−0.07	**<0.001**
agreeableness		−0.0003	[−0.004 0.003]	−0.002	0.84
**conscientiousness**		−0.007	[−0.009 −0.006]	−0.08	**<0.001**
**CES-D—sample 1 (df = 8938)**					
**Model**	0.04				**<0.001**
**sex**		0.04	[0.032 0.055]	0.08	**<0.001**
**education**		−0.09	[−0.10 −0.08]	−0.20	**<0.001**
**age**		0.001	[0.0007 0.0016]	0.05	**<0.001**
**secondary DRS**		0.002	[0.0003 0.003]	−0.03	**0.02**
**CES-D—sample 2 (df = 7896), corrected for personality**					
**Model**	0.21				<0.001
**sex**		0.013	[0.0010 0.0261]	0.02	**0.05**
**education**		−0.06	[−0.068 −0.048]	−0.12	**<0.001**
**age**		0.0006	[0.0001 0.0011]	0.03	**0.01**
secondary DRS		0.001	[−0.005 0.002]	0.01	0.20
**neuroticism**		0.024	[0.022 0.025]	0.36	**<0.001**
**extraversion**		−0.006	[−0.008 −0.005]	−0.08	**<0.001**
**openness**		−0.007	[−0.010 −0.005]	−0.07	**<0.001**
agreeableness		−0.0003	[−0.004 0.003]	−0.002	0.84
**conscientiousness**		−0.008	[−0.009 −0.006]	−0.08	**<0.001**

Abbreviations: CES-D = depressive symptoms score, DRS = dietary restriction score. Significant associations (p-values < 0.05) are indicated in bold.

**Table 8 nutrients-12-01492-t008:** MANCOVA analysis of dietary restriction, age, sex, education on personality (*n* = 7906).

	Pillai’s Trace	F	df	num df	den df	p
**NEOFFI (all factors)—sample 2, corrected for age, sex, education**						
sex	0.169	320.0	1	5	7897	<0.001
education	0.041	67.4	1	5	7897	<0.001
age	0.040	65.2	1	5	7897	<0.001
overall DRS	0.007	11.8	1	5	7897	<0.001
**NEOFFI Neuroticism**						
**sex**		342.0	1	5	7897	**<0.001**
**education**		114.5	1	5	7897	**<0.001**
**age**		28.9	1	5	7897	**<0.001**
overall DRS		0.6	1	5	7897	0.44
**NEOFFI Extraversion**						
**sex**		14.5	1	5	7897	**<0.001**
**education**		72.6	1	5	7897	**<0.001**
**age**		149.3	1	5	7897	**<0.001**
overall DRS		0.3	1	5	7897	0.6
**NEOFFI Openness**						
**sex**		6.1	1	5	7897	**0.01**
**education**		209.8	1	5	7897	**<0.001**
**age**		4.9	1	5	7897	**0.03**
overall DRS		1.6	1	5	7897	0.21
**NEOFFI Agreeableness**						
**sex**		937.3	1	5	7897	**<0.001**
education		0.9	1	5	7897	0.34
age		0.2	1	5	7897	0.7
**overall DRS**		15.7	1	5	7897	**<0.001**
**NEOFFI Conscientiousness**						
**sex**		122.4	1	5	7897	**<0.001**
**education**		10.7	1	5	7897	**0.001**
**age**		130.7	1	5	7897	**<0.001**
**overall DRS**		53.9	1	5	7897	**<0.001**

Abbreviations: DRS = dietary restriction score. Significant associations (p-values < 0.05) are indicated in bold.

**Table 9 nutrients-12-01492-t009:** Multiple regression analyses predicting CES-D as a function of age, sex, education and dietary restriction score (sample 1 *n* = 8943, sample 2 *n* = 7906).

	Adj. R^2^	B	C.I.	Beta	*p*
**CES-D—sample 1 (df = 8938)**					
**Model**	0.05				**<0.001**
**sex**		0.04	[0.032 0.055]	0.076	**<0.001**
**education**		−0.09	[−0.100 −0.080]	−0.185	**<0.001**
**age**		0.001	[0.0008 0.0017]	0.054	**<0.001**
**overall DRS**		−0.004	[−0.006 −0.002]	−0.043	**<0.001**
**CES-D—sample 2 (df = 7901)**					
**Model**	0.04				<0.001
**sex**		0.04	[0.031 0.056]	0.075	**<0.001**
**education**		−0.09	[−0.100 −0.080]	−0.180	**<0.001**
**age**		0.001	[0.0008 0.0017]	0.054	**<0.001**
**overall DRS**		−0.005	[−0.007 −0.002]	−0.048	**<0.001**
**CES-D—sample 2 (df = 7896), corrected for personality**					
**Model**	0.21				<0.001
**sex**		0.014	[0.0010 0.0261]	0.02	**0.04**
**education**		−0.06	[−0.068 −0.048]	−0.12	**<0.001**
**age**		0.0007	[0.0002 0.0011]	0.03	**0.007**
**overall DRS**		−0.003	[−0.004 −0.001]	−0.03	**0.007**
**neuroticism**		0.024	[0.022 0.025]	0.36	**<0.001**
**extraversion**		−0.006	[−0.008 −0.005]	−0.08	**<0.001**
**openness**		−0.007	[−0.010 −0.005]	−0.07	**<0.001**
agreeableness		−0.0005	[−0.004 0.003]	−0.004	0.76
**conscientiousness**		−0.007	[−0.009 −0.006]	−0.08	**<0.001**

B/beta represent unstandardized/standardized regression coefficients. Abbreviations: CES-D = depressive symptoms, DRS = dietary restriction score. Significant associations (p-values < 0.05) are indicated in bold.

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
