# Peer review of "Less Animal-Based Food, Better Weight Status: Associations of the Restriction of Animal-Based Product Intake with Body-Mass-Index, Depressive Symptoms and Personality in the General Population"

_nutrients, 2020, doi:10.3390/nu12051492_

Round 1

Reviewer 1 Report

Thank you to the editor and the authors for the opportunity to review this manuscript. The authors verify how exclusion of animal-based food is associated with BMI, personality traits and symptoms of depression.

In the introduction strong arguments were made for designing this study. In my opinion, the methodology and analysis of the results is statistically sound. The discussion is well-formulated with an in-depth analysis of obtained results. The authors in the discussion referred to the results of other studies and showed what gap in our knowledge they filled. They also refer to the latest research on the gut-brain axis. Noteworthy is the pre-registration of the research and making the script to the R available.

Basically, I do not have any major question on this manuscript. However, I have a minor recommendations:

1) In the graphic abstract it is incomprehensible which picture shows which diet. Please sign the pictures, because "diet" alone is not enough.

2) In the limitation section, it should be added that the study verifies depressive mood or depressive symptoms and not depression. The clinical interview remains the gold standard for identifying depression.

Overall, this scientifically-sound manuscript will serve as an important research contribution to the field of diet influence on mood.

Author Response

Response to Reviewer 1 Comments

We thank the reviewers for the possibility to revise the manuscript and their encouraging remarks. Implementing the feedback we could make a substantial improvement of the manuscript possible.

Point 1: In the graphic abstract it is incomprehensible which picture shows which diet. Please sign the pictures, because "diet" alone is not enough.

Response 1: The descriptive terms in the graphical abstract were updated as requested to be more distinct into “animal-origin restrictive diet” and “overall restrictive diet”.

Point 2: In the limitation section, it should be added that the study verifies depressive mood or depressive symptoms and not depression. The clinical interview remains the gold standard for identifying depression.

Response 2: Thank you for pointing this out. Wording has been adapted accordingly to “depressive mood and depressive symptoms” in the Limitation Section and added the following sentence (changes in bold):

“Firstly, limitations of our study include that the results are based on a cross-sectional study design and therefore cannot explain underlying causalities.

Secondly, our analyses are based on self-reported dietary food record, which do not necessarily reflect actual food intake, however, test-retest reliability is generally of good quality [65]. Moreover, the FFQ used did not ask for quantity of food intake, which limits the interpretability of the observed effects (for further discussion on possible mechanisms see [1]). Yet, beside this possible inaccuracy of self-reported food intake, we propose that excluding certain food groups for a timeframe of 12 months presumably is a strong and reliable indicator of actual food intake and exclusion of certain food groups.

Thirdly, this study verifies depressive mood or depressive symptoms and not depression. The clinical interview remains the gold standard for identifying depression.”

Reviewer 2 Report

  1. Abstract and Intro need grammar editing.
  2. Is there a more universal term that can be used in place of wurst? Sausage?
  3.  Add additional information about the Life-Adult cohort in materials and methods. Data collected over multiple years? Location? Ages included? What are the Adult Baseline and Adult Baseline Plus samples?
  4. Figure 1: subtracting exclusion criteria from total n of 10448 leaves me with a final sample 1 of 8940
  5.  Why did the authors not include additional demographic information related to diet such as race, ethnicity, and income?
  6.  Please describe the scoring for the NEOFFI-30 and CES-D. What does a low vs high number for each indicate?
  7.  Was a validated FFQ used? If so please add a references. If not, this should be indicated in the limitations. 
  8.  Please disuss the data in Table 1. Are the results representative of the population? 
  9. Please discuss the implications of demographic factors on future research directions. 
  10.  Did the authors consider total kcal intake in any of the analyses? Animal products as a percentage of total intake may lead to additional information about the relationships investigated.

Author Response

Response to Reviewer 2 Comments

We thank the reviewers for the possibility to revise the manuscript and their encouraging remarks. Implementing the feedback we could make a substantial improvement of the manuscript possible.

Point 1: Abstract and Intro need grammar editing.

Response 1: Thank you for pointing this out. We revised the whole manuscript carefully for correct grammar use, please see edits in the text (Track Changes) and sample examples here (in green).

“Abstract

[...] Moreover, various personality traits… [...] We aim to systematically assess [...]

Introduction

[...] diets are associated with lower all-cause mortality and less frequently with cardiovascular diseases [2,3].

[...] however showed no effect of a plant-based diet on mortality rate. The 18-year follow-up [...]

[...] Intervention studies in small to moderate sample sizes (n∼100) indicated that medium-term (12-74 weeks) vegan diets, compared to omnivorous diets, led to weight loss [...]”

Point 2: Is there a more universal term that can be used in place of wurst? Sausage?

Response 2: We agree with the Reviewer that the term 'wurst' might be too specific. In German-speaking cultures, wurst is defined as processed meat, including sausages but also cold cuts, the latter usually put upon slices of bread or rolls. For better comprehension among the scientific community, we replaced all mentions of wurst in the text with “cold cuts”.

Point 3: Add additional information about the Life-Adult cohort in materials and methods. Data collected over multiple years? Location? Ages included? What are the Adult Baseline and Adult Baseline Plus samples?

Response 3: We followed the Reviewer's request and added more details about the study with the following information to the Materials and Methods part of the manuscript (shown in bold):

“[...] Participants were drawn from the population-based Leipzig Research Centre for Civilization Diseases (LIFE)-Adult cohort, which aims to explore causes and developments of common civilization diseases such as obesity, depression and dementia (see [28] for details and Figure 1). Briefly, n > 9,500 adult participants ("Adult Baseline") were randomly selected based on sex and year of birth (age range 18-80 y, with a main proportion focus on 40-80 y), from the city registry of Leipzig, Germany, a major city with 550,000 inhabitants in the east of Germany. Additional volunteers (n > 900, randomly recruited from the city registry and from local databases, "Adult Baseline Plus") were included from periods of feasibility testing, piloting and finalization. Data collection was conducted from August 2011 to November 2014 at a single site in cooperation of the Faculty of Medicine, Leipzig University and the Max Planck Institute for Human Cognitive and Brain Sciences. All participants underwent anthropometric measurements and answered extensive questionnaires regarding dietary habits, depressive mood and personality (see below for details).

2.1. Inclusion criteria.

The initial dataset consisted of n = 10,083 participants. Subjects were included into the analysis based on standardized rules [...]”

Point 4: Figure 1: subtracting exclusion criteria from total n of 10448 leaves me with a final sample 1 of 8940

Response 4: We thank the Reviewer for bringing this to our attention. We indeed counted the missings for height twice, and thus revised Figure 1 so that it now shows the correct numbers. Also we included the specification of the samples as described above for Baseline and Baseline Plus subsamples.

Point 5: Why did the authors not include additional demographic information related to diet such as race, ethnicity, and income?

Response 5: We did not report race and ethnicity, as this information was not available for the cohort. Germany usually does not collect racial statistics. Considering overall ethnicity distribution in Germany though, it has been estimated that the vast majority belongs to the same ethnic group, about ~1% of the population are African-Germans; nothing can be reliably guessed about East Asians, Americans or others. The proportion of residents with a migration background is smaller in old vs. young, and in Eastern compared to Western Germany (8% vs 28%), with the city of Leipzig having 9.3% in 2012. Most of those with migration history in Leipzig stem from Turkey, Poland, and former Soviet states, the latter comprising mostly late German repatriates of World War II ("Spätaussiedler"). In the former German Democratic Republic (DDR, Leipzig was part of it until it decades 1989), migration included workers from Poland and Hungary, later Cuba, Mozambique and Vietnam, however migration in total was very limited in the DDR. This leads us to conclude that the majority of the Leipzig cohort were probably of Caucasian ethnicity, and <1% were non-Caucasian. We added the lack of information considering ethnicity as a limitation to the discussion part, where it now reads:

“Fourthly, we did not account for ethnicity as these statistics were not available for this cohort. We suppose the majority of the LIFE cohort were probably of Caucasian ethnicity and <1% were non-Caucasian, due to historical data on immigration for this region.

Lastly, as frequently done in nutritional epidemiology, in our analysis socioeconomic status was accounted for by level of education, not income and occupational status (Temple, 2015). This one-dimensional analysis might result in limited generalizability of the results. However, education can be viewed as a more long-term indicator of socioeconomic status compared to more dynamic monthly net income (as provided in this dataset).”

Temple, N. (2015). The possible importance of income and education as covariates in cohort studies that investigate the relationship between diet and disease. F1000Research, 4.

Point 6: Please describe the scoring for the NEOFFI-30 and CES-D. What does a low vs high number for each indicate?

Response 6: We have added the following description to the respective Methods Section (see also analysis code available at OSF):

“2.5. Personality.

Personality traits were measured with the German version of the Big Five via Short Forms (16-Adjective Measure) [31]; subscales were computed for Neuroticism, Extraversion, Openness, Agreeableness and Conscientiousness by building summed scores according to the test’s manual (higher scores indicate more pronounced traits, lower scores indicate less pronounced traits). In a subsample personality traits were measured with the German version of the NEOFFI-30 [32,33].

“2.6. Depressive scores.

Depressive scores (self-reported) were assessed by the Centre of Epidemiologic Studies-Depression (CES-D) scale [34]. Total CES-D score was calculated as a sum of responses to all 20 questions, with higher scores indicating more diverse and/or more frequent depressive symptoms.

Point 7: Was a validated FFQ used? If so please add a references. If not, this should be indicated in the limitations. 

Response 7: We agree with the Reviewer that a lack of validation of the FFQ should be noted as limitation. The FFQ was created based on different questionnaires available in German language of previous epidemiological studies, however of unknown source. We added the following section in the limitations (shown in bold):

“[...] Secondly, our analyses are based on self-reported dietary food record, which do not necessarily reflect actual food intake, however, test-retest reliability is generally of good quality [65]. The FFQ used in this study has been adopted from the German National Health Interview and Examination Survey 1998 (Finger et al., 2013) lacking proper validation. [...]”

Finger, J. D., Tylleskär, T., Lampert, T., & Mensink, G. B. (2013). Dietary behaviour and socioeconomic position: the role of physical activity patterns. PloS one, 8(11).

Point 8: Please disuss the data in Table 1. Are the results representative of the population? 

Response 8: We agree that the sample characteristics are worth to be discussed in the light of representation of the general population. We added the following to the manuscript (shown in bold):

“3. Results

We included 8,943 subjects for analyses regarding diet, BMI and depressive symptoms (see Table 1 for demographics), and 7,906 participants in sample 2 for the subsample analysis additionally investigating personality traits (see Table 2). Due to the focus on the age range between 40-80 years in the randomized selection of participants drawn from the city registry (details see [28]), the studied sample was on average middle-aged (mean age 57y) and showed a skewed adult age distribution to the right. In addition, the study included slightly more women than men (4609F, 4334M) and a very wide BMI range (16-57 kg/m2, on average 27 kg/m2)."

“4. Discussion

[...] Surprisingly, not diet but personality was significantly associated with depression scores. Based on the randomized selection of residents of Leipzig, our findings could overall generalize to a mid- to older adult rural population of Eastern Germany, with one common limitation that populations with a low social status and an unhealthy lifestyle might be somewhat underrepresented (Enzenbach et al. 2019). The BMI range can be considered representative of a German population of this age.”

Enzenbach, C., Wicklein, B., Wirkner, K., & Loeffler, M. (2019). Evaluating selection bias in a population-based cohort study with low baseline participation: the LIFE-Adult-Study. BMC medical research methodology, 19(1), 135.

Point 9:  Please discuss the implications of demographic factors on future research directions. 

Response 9: Following the reviewer's request, we added  the following to the Discussion Section:

“[...] Our results further highlight a significant association of demographic variables with BMI, personality and depressive symptoms. This shows how individual factors such as age, sex and education are tightly linked to health (dis)advantageous in our societies, and future studies on public health interventions should focus on those at particular risk, e.g. older males with lower education in the case of BMI, and older females with lower education in the case of depressive symptoms. In parallel, the cross-sectional nature of our analysis does not allow to imply causal relations, therefore future longitudinal and experimental interventional studies need to test to what extent modifiable factors, such as education, could causally reduce obesity and depression, and how dietary strategies such as reducing animal-based products might help to mediate these effects.”

Point 10: Did the authors consider total kcal intake in any of the analyses? Animal products as a percentage of total intake may lead to additional information about the relationships investigated.

Response 10: Thank you for this very important remark, as total kcal intake is an important factor to consider in dietary intake analyses. As the used FFQ only asked for frequency over the last 12 months and not quantity of intake, we could not calculate total kcal intake. We added this as a limitation, where it reads:

“Secondly, our analyses are based on self-reported dietary food record, which do not necessarily reflect actual food intake, however, test-retest reliability is generally of good quality [65]. Moreover, the FFQ used did not ask for quantity of food intake, which limits the interpretability of the observed effects, as important confounders such as total kcal intake could not be considered in this analysis (for further discussion on possible mechanisms see [1]).”
